# THINKING LIKE A CHEMIST: COMBINED HETEROGENEOUS EMBEDDING MODEL INTEGRATING STRUCTURE AND TOKENS

## ABSTRACT

Representing molecular structures effectively in chemistry remains a challenging task. Language models and graph-based models are extensively utilized within this domain, consistently achieving state-of-the-art results across an array of tasks. However, the prevailing practice of representing chemical compounds in the SMILES format – used by most data sets and many language models – presents notable limitations as a training data format. In this study, we present a novel approach that decomposes molecules into substructures and computes descriptor-based representations for these fragments, providing more detailed and chemically relevant input for model training. We use this substructure and descriptor data as input for language model and also propose a bimodal architecture that integrates this language model with graph-based models. As LM we use RoBERTa, Graph Isomorphism Networks (GIN), Graph Convolutional Networks (GCN) and Graphormer as graph ones. Our framework shows notable improvements over traditional methods in various tasks such as Quantitative Structure-Activity Relationship (QSAR) prediction.

## 1 INTRODUCTION

The integration of machine learning (ML) has emerged as a transformative force in the natural sciences, particularly in the discipline of chemistry (Chithrananda et al., 2020; Hu et al., 2016; Wang et al., 2022). This integration encompasses various tasks, ranging from the regression of molecular properties, exemplified by quantitative structure-activity relationship (QSAR) models (Wu et al., 2021; Rácz et al., 2021), to complex challenges, such as predicting nuclear magnetic resonance (NMR) spectra from the structure of chemical compounds (Yao et al., 2023). As an ever-evolving discipline, the latest advancements in machine learning are gradually being adapted for applications in chemistry, albeit with some delay. Molecular representations are fundamental to the application of machine learning in chemistry, and three primary types are typically employed: graph-based (Reiser et al., 2022), string-based (Heller et al., 2015; Weininger, 1988; Krenn et al., 2020), and vector representations (Rogers & Hahn, 2010; Durant et al., 2002).

Graph-based representations conceptualize chemical compounds as molecular graphs, effectively capturing their structural properties (David et al., 2020; Kwon et al., 2020). This format naturally aligns with graph neural networks, which have been successfully applied to numerous chemical problems, demonstrating their efficacy in molecular analysis. String representations, particularly Simplified Molecular Input Line Entry System (SMILES) (Weininger, 1988), are widely regarded as a standard method for the linear representation of molecular structures. SMILES is typically used for storing compounds in databases and, despite its limitations, effectively represents the structure of a molecule (Chithrananda et al., 2020; Ahmad et al., 2022; Wang et al., 2019; Shamshad et al., 2023; Cong et al., 2024) and serves as a basic data representation for language models (Chithrananda et al., 2020; Ahmad et al., 2022; Ross et al., 2022). However, it presents notable shortcomings (Ganeeva et al., 2024). Initially designed for efficient storage and representation of molecular data, SMILES lacks comprehensive information regarding the physical and chemical properties of compounds. Two main approaches exist to overcome these difficulties. The first combines graph models with SMILES-based natural language processing (NLP) transformers (Zhu et al., 2023), integrating the strengths of both methodologies.

The second is based on changing the representations of molecules used in training. In cheminformatics, molecules are commonly represented as vectors in a high-dimensional space, where each vector encodes essential molecular features to enable effective computational analysis. Molecular descriptors (Sahoo et al., 2016; Bhatia et al., 2023; Darlami & Sharma, 2024) – numerical values summarizing properties such as size, shape, and electronic characteristics – are central to this representation, providing critical information for predicting molecular behavior and interactions. Traditional global descriptors capture information from the entire molecule but often fail to reflect important structural variations within specific subregions. To address this limitation, we propose focusing on descriptors derived from molecular substructures, offering a more detailed and accurate representation that better captures the complexity of molecular behavior. There are existing attempts to use quantum-chemical descriptors as a basis for transformer-based models (Zhou et al., 2023).

Thus, there is a legitimate idea of combining these two approaches: adapting descriptors as a data format for language models and applying bimodal architectures.

**Contributions.** We propose an approach that involves decomposing molecules into chemically meaningful substructures and calculating descriptors for each segment. These sequences of substructure descriptors serve as input to the RoBERTa model (Liu, 2019), enabling it to learn and capture the underlying physicochemical relationships within the molecule. This helps in improving performance in tasks such as predicting molecular properties (Wu et al., 2021; Rácz et al., 2021; Sabando et al., 2022), classifying biological activity, generating novel molecules, and studying molecular interactions (Hu et al., 2016).

The graph-based model provides a detailed representation of molecular structures, making it especially effective for analyzing large compounds with multiple substructures. By explicitly capturing the connectivity and spatial relationships among substructures, it overcomes the limitations of language models that lack detailed molecular organization. To improve structural representation and property prediction, we use atom masking and graph augmentation with Graphormer, an advanced architecture designed to capture complex relational patterns in molecular graphs.

We propose two bimodal architectures combining RoBERTa with a graph convolutional network (GCN) (Kipf & Welling, 2016) and a Graph Isomorphism Network (GIN) (Xu et al., 2018), using contrastive learning to enhance feature extraction. Although these models train faster, they may underperform on complex tasks compared to a more advanced system integrating RoBERTa with Graphormer (Ying et al., 2021), which specializes in modeling intricate relations. In all graph models, we apply masking of atom and edge features during training to predict masked elements and align embeddings of augmented molecular views. This contrastive learning approach is underexplored in cheminformatics and shows promise for advancing molecular modeling. Code is available G.

## 2 RELATED WORK

**Molecular descriptors.** The relationships between molecular properties have been extensively studied and form a foundation for property prediction in chemoinformatics (Hansch et al., 1962; 1963; Hansch & Fujita, 1964). Molecular descriptors, which quantitatively represent molecular structure and characteristics, encompass various types such as substructural descriptors (e.g., MACCS keys), topological descriptors derived from 2D molecular graphs, and geometric descriptors capturing 3D molecular shape and spatial configuration. These descriptors serve as key inputs for QSAR/QSPR models that aim to predict molecular properties from structural information (section A). While conventional models typically rely on whole-molecule descriptors, they often overlook the nuanced influence of individual substructures, particularly in larger, multifunctional molecules (Estrada, 2008). To address this limitation, recent strategies focus on fragment-based descriptors, enabling more localized and interpretable predictions.

When applied as representations for language models, molecular descriptors offer promising advantages. Certain descriptors encode positional information reminiscent of natural language sequences while simultaneously reflecting important physicochemical properties. This dual capability allows language models to capture both statistical patterns and chemically meaningful relationships, improving their ability to generate insightful and reliable predictions.

**BRICS fragmentation.** The Breaking of Retrosynthetically Interesting Chemical Substructures (BRICS, section B) method (Degen et al., 2008) offers a principled, rule-based approach to fragmenting molecules into chemically meaningful components by selectively cleaving bonds commonly involved in synthesis, guided by 16 well-defined rules. This ensures that the resulting fragments correspond to synthetically relevant building blocks while retaining information about their potential attachment points, facilitating their recombination in silico. Widely adopted in drug discovery and computational chemistry, BRICS mirrors the logic of retrosynthetic analysis by breaking down complex molecules into simpler precursors.

In our work, we utilize BRICS to preprocess data by partitioning molecules into fragments and computing descriptors for each. This fragment-centric strategy shifts focus from the whole molecule to its local chemical environments, improving the model's ability to predict properties influenced by specific substructures and enabling a more interpretable analysis of structure–property relationships.

**SMILES-based NLP models.** Transformers (Vaswani, 2017) were initially introduced to facilitate the generation of vector representations for natural language processing tasks. Since their inception, they have found widespread application across various domains, including speech recognition, medicine, and neuroscience (Shamshad et al., 2023; Cong et al., 2024).

There have been several efforts to adapt transformers for chemical applications, exemplified by models such as SmilesBERT (Wang et al., 2019), ChemBERTa (Chithrananda et al., 2020), and ChemBERTa-2 (Ahmad et al., 2022), and almost all of them were trained on SMILES. Many of these models have been trained on substantial datasets, including ZINC (Irwin et al., 2012) and PubChem (Kim et al., 2023), demonstrating commendable performance in classification and regression tasks across various established chemical benchmarks.

**Graph models.** Graph neural networks (GNNs) have effectively addressed a variety of challenges within the field of chemistry (David et al., 2020; Kwon et al., 2020). Many GNNs are highly specialized for specific tasks and are not inherently designed to generate vector representations of chemical compounds.

Several methodologies have been proposed to enhance GNN-based embeddings. For instance, (Hu et al., 2016) introduced two primary concepts: the recovery of masked properties of a molecule, such as the type of a specific atom, and the application of contrastive learning to minimize discrepancies between two subgraphs within a molecule. Additionally, Mol-CLR (Wang et al., 2022) presents a framework based on the augmentation of molecular graphs through the removal of atoms, edges, and subgraphs, followed by the training of a model to reconstruct these components. However, many GNNs are specialized for specific tasks and are not inherently designed to generate vector representations of chemical compounds.

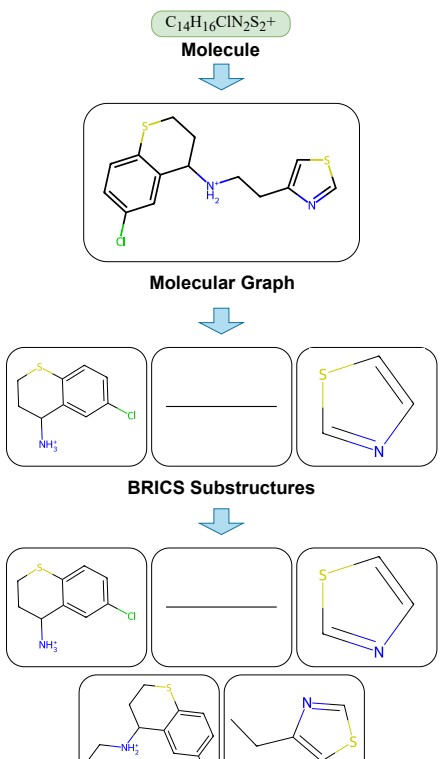

Figure 1: An example of splitting molecule to substructures and creating the additional substructure (in case of serotonin, its the parent molecule due to it was splitted to only two BRICS blocks).

In the graph component of our model, we advocate for an approach that synthesizes these concepts and leverages state-of-the-art models. Specifically, we implement a mechanism to mask atom features and edge features in the case of Graphormer (Ying et al., 2021). The model is trained not only to predict these masked features but also to align the embeddings of two augmented versions of the same molecule. This approach represents a modification of contrastive learning, a technique that remains underutilized in the chemistry domain. Moreover, (Zhu et al., 2023) introduced a bimodal architecture incorporating a BERT-based language model (LM) trained on SMILES alongside a

GNN as the graphical representation model. In contrast, we propose a distinct language model that is trained on fingerprints, thus providing a more physically informed perspective and an advanced graph model. Additionally, our approach includes notable differences in the final projection and the processing of embeddings derived from both the language and graph models.

# 3 DATA PREPROCESSING

We propose a data preparation methodology for language models that focuses on representing local substructural properties through physicochemical descriptors of molecules. The process divides into several key stages. First, molecules represented by SMILES are partitioned into substructures using the BRICS algorithm. Then, additional substructures are generated (the comparison of models trained with and without additional substructures is presented in Appendix C): for every bond removed during this fragmentation, a new substructure is created that comprises the two BRICS-derived substructures originally connected by that bond (see Fig. 1). To maintain chemical completeness, all substructures are augmented with hydrogen atoms according to valence rules, compensating for broken bonds in the parent molecule. This design, somewhat analogous to circular fingerprints such as ECFP (Rogers & Hahn, 2010) or Morgan fingerprints (Morgan, 1965), enables the model to capture information not only about individual substructures but also about the connections between them.

Subsequently, a set of structural, topological, and physicochemical descriptors (described in the appendix A) is calculated for each substructure via RD-Kit. These descriptors are organized into an ordered array, which is crucial for incorporating positional information and the usage of positional encoding in RoBERTa). The selected descriptors capture various aspects, including the topological structure of the molecule, the number of atoms in each substructure, as well as certain quantum and physicochemical properties (see the first two arrows in Fig. 5). As a result, each molecule is transformed first into a collection of relevant substructures and is represented as a two-dimensional array, where each row corresponds to the descriptor vector of an individual substructure (see the next section).

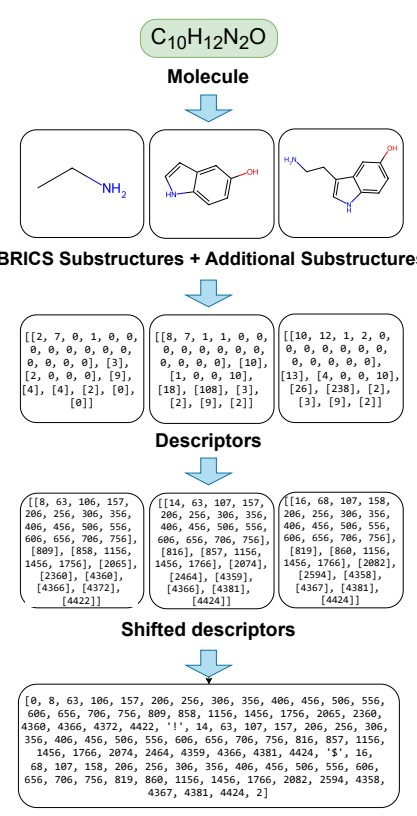

Figure 2: Example of tokenization process. Toketns "0" and "2" correspond to BOS (begin of sequence) and EOS (end of sequence), respectively. The '!' and '$' kept non-tokenizenised for clarity.

**Discussion.** Our approach relies on partitioning molecules into chemically meaningful substructures using the BRICS method, rather than applying conventional BPE-like tokenization, which often emphasizes frequently occurring but not necessarily chemically relevant fragments. Importantly, the BRICS fragmentation is guided by a set of rules specifically developed for retrosynthetic analysis, mirroring the way chemists cognitively organize molecular structures into building blocks when designing new compounds or predicting their properties. Since chemistry as a scientific discipline is fundamentally rooted in human reasoning and intuition, enabling the model to understand molecules through these chemically intuitive substructures allows it to capture the underlying chemical principles more effectively. By aligning the model's perspective with the way chemists think, rather than relying solely on statistical patterns, we improve its ability to grasp the chemical laws and concepts established through decades of human expertise, thereby enhancing both interpretability and practical relevance.

It is also important to recognize that many molecular properties cannot be accurately represented as a simple linear combination of the properties of individual substructure descriptors. This challenge is

analogous to the problem of constructing a meaningful sentence embedding from the embeddings of its constituent words. In this context, substructures can be viewed as words, while the entire molecule corresponds to a sentence, with the number of substructures varying between molecules. The RoBERTa architecture proves to be particularly well-suited for addressing this type of variable-length, context-dependent representation, making it an optimal choice for modeling molecular properties in this framework. Moreover some compounds (e.g., metal complexes or boron clusters) cannot be represented in SMILES, but a molecular graph and descriptors can be generated for them. Our approach can handle these compounds, whereas SMILES-based models cannot at all.

# 4 ARCHITECTURE AND TRAINING DETAILS

## 4.1 ARCHITECTURE OVERVIEW

The proposed model comprises three primary components, as illustrated in Figure 3: the graph model, the language model, and the projection blocks. The language model is designed to accept two-dimensional arrays of substructure descriptors as input, whereas the graph model processes molecular graphs. The function of the projection blocks is to transform the embeddings generated by the graph and language models from their respective latent spaces into a unified third latent space. The baseline of our approach is the language model itself (without graph parts), and as shown in Section 5, it outperforms other modern frameworks in most benchmarks.

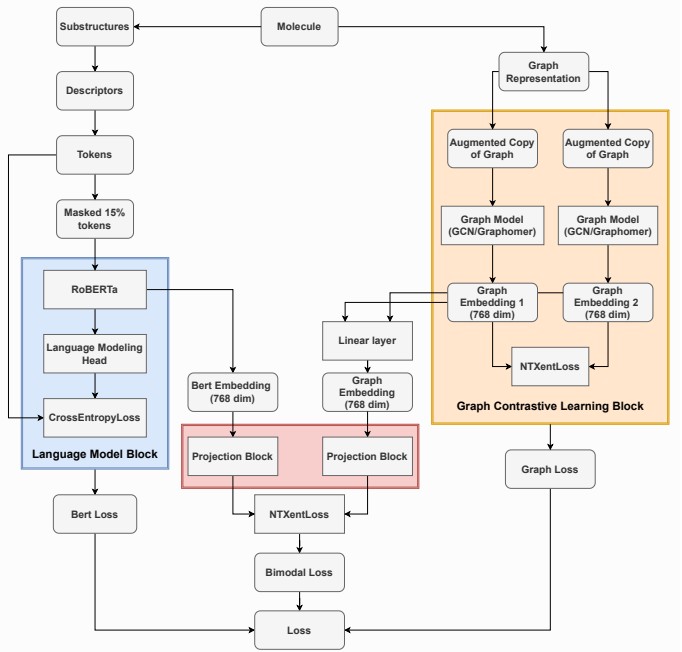

Figure 3: Full architecture of bimodal model. Language and Graph blocks are outlined by blue and orange colors. Red color marks projection blocks.

## 4.2 LANGUAGE MODEL

**Tokenizer.** Our input to the language model consists of two-dimensional arrays of descriptors, where each array represents a single molecule and each subarray corresponds to a specific substructure within that molecule. The tokenization process applied to these arrays is relatively straightforward, as the nature of the chosen descriptors eliminates the need for conventional statistical tokenization methods, such as Byte Pair Encoding (BPE), which aim to adjust the dictionary size.

Instead, the tokenization involves four key steps. First, within each subarray, the descriptor at position $i$ is adjusted by adding the sum of the maximum possible descriptor values from the preceding positions $[0, i-1]$. This operation effectively tokenizes the descriptor values in a position-aware manner (see the last two arrows on Fig. 2). Next, a special separator token, denoted here as '!', is inserted between descriptor subarrays to clearly signal the boundary between different substructures. To further highlight the chemical significance of the fragmentation, an additional token '$' is inserted to separate traditional BRICS-derived substructures from the so-called additional substructures (as detailed in the Data Preprocessing). Finally, the resulting two-dimensional array is flattened into a one-dimensional sequence, with tokens indicating the start and end of the sequence appended at the respective positions.

**RoBERTa training.** We utilize the RoBERTa architecture (Liu, 2019), which has been trained on descriptors for molecules derived from the PubChem (Kim et al., 2023) dataset, as our language model. Within this framework, the encoding of an individual descriptor is interpreted as a "word", while the substructure in a molecule is interpreted as a "sentence" and lastly the encoding of an entire molecule is considered analogous to "text". During the training process, the 1-dimensional array obtained after the tokenization process undergoes standard procedures, which was given

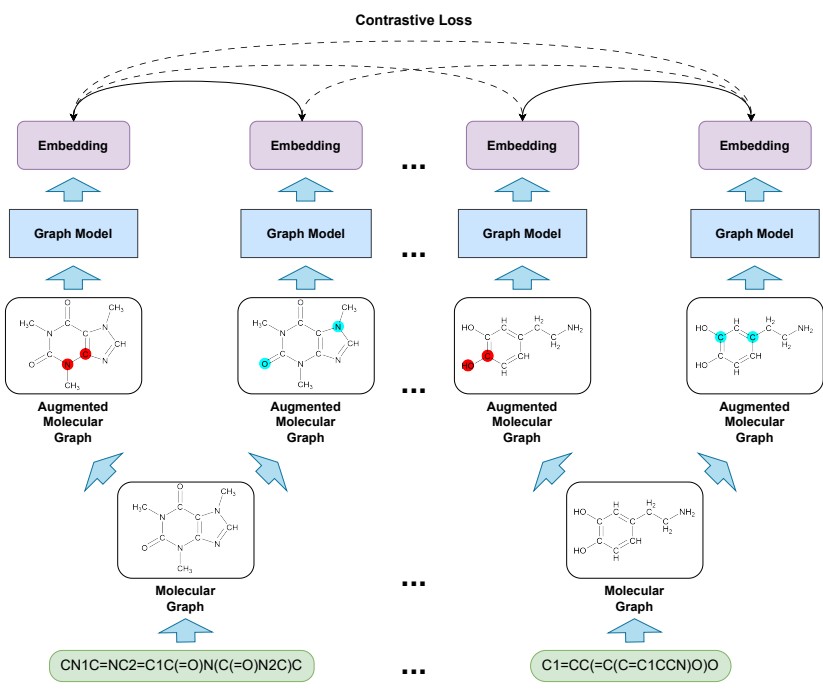

Figure 4: Tops masking process and computing the graph loss for one batch.

after tokenization process, undergoes standard procedures including the masking of 15% of tokens (representing descriptors), with the model subsequently predicting the probabilities of these masked tokens. The output embedding is derived from the CLS token located in the penultimate layer of the model.

## 4.3 GRAPH MODEL

**Creation and augmentation of graph.** A graph is constructed from SMILES representations utilizing the RDKit package, wherein each atom is represented as a vertex. Two parameters – atom number and chirality – are designated as attributes of the vertices. In this framework, each bond is represented as an edge, with the bond multiplicity (single, double, triple, or aromatic) serving as the attribute for the edges.

Subsequently, 20% of the atomic attributes are masked, replacing them with a designated mask token. In the case of graphomers, an equivalent approach is applied, where 20% of the edge attributes are also masked, transforming these attributes into the mask token. The augmentation process and the graph model operation scheme are shown in Figure 4.

**Model training.** In the graph component of our model, we have experimented with three distinct architectures: Graph Isomorphism Network (GIN), Graph Convolutional Network (GCN), and Graphormer. We employ augmentation techniques to transform the molecular graph into two distinct representations. Following this, we train the GCN, GIN, or Graphormer models with the objective of minimizing the differences between the augmentations of one graph and maximizing the differences between augmentations of different graphs (this process for graphs in one batch is shown in Figure 4).

## 4.4 CONNECTION BETWEEN MODELS

The projection blocks illustrated in Figure 5 of our proposed architecture comprise two linear layers accompanied by two batch normalization blocks. Prior to the application of the final batch normalization block, the ReLU activation function is employed on the embeddings.

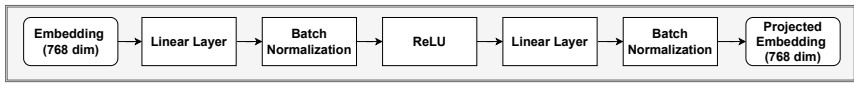

**Projection Block**

Figure 5: The structure of the projection block. It helps to translate output vectors from models to the same linear space.

Let $e_{\text{graph}}$ denote the output of the graph model and $e_{\text{lang}}$ represent the output of the language model. Furthermore, let $\psi_{\text{graph}}$ and $\psi_{\text{lang}}$ be the respective projection blocks for the graph and language models. Define $\mathbb{A}$ as the latent space of the graph model, $\mathbb{B}$ as the latent space of the language model, and $\mathbb{C}$ as the space into which the embeddings are projected. Thus, we have $e_{\text{graph}} \in \mathbb{A}$, $e_{\text{lang}} \in \mathbb{B}$ with $\psi_{\text{graph}} : \mathbb{A} \to \mathbb{C}$ and $\psi_{\text{lang}} : \mathbb{B} \to \mathbb{C}$.

### 4.5 LOSS FUNCTIONS

The loss function used in our model is represented as

$$L = \alpha \cdot L_{\text{lang}} + \beta \cdot L_{\text{graph}} + \gamma \cdot L_{\text{bimodal}}, \tag{1}$$

where $L_{\text{lang}}$ is the loss function of the language model, $L_{\text{graph}}$ is the loss function of the graph part of the model, and $L_{\text{bimodal}}$ is the embedding projection loss function from the graph and language models. The coefficients $\alpha$, $\beta$, and $\gamma$ are constants that can be considered hyperparameters and are assigned default values of $1.0$.

**Language model loss.** $L_{\text{lang}}$ is calculated as the regular Cross-Entropy applied to the labels and predicted tokens of the language model.

**Graph model loss.** $L_{\text{graph}}$ is defined as NTXent-Loss (Sohn, 2016) applied to the batch of augmented graphs' embeddings and to the batch of original graphs' embeddings. It tries to minimize the distance between augmented and original embeddings of the same index and distances others with different indices. NTXent-Loss calculates the cosine distance between two vectors and utilizes the temperature parameter to balance the influence of positive and negative pairs. Let $sim(u, v)$ denotes the cosine similarity between vectors $u$ and $v$. Then, the loss function for a positive pair of examples (i, j) is as follows:

$$(L_{\text{graph}})_{i,j} = -\log \left( \frac{e^{\text{sim}(u_i, v_j)/\tau}}{\sum_{k=1}^{N} e^{\text{sim}(u_i, v_k)/\tau}} \right), \tag{2}$$

where $N$ is the total number of examples and $\tau$ (temperature) is a parameter that controls the contribution of positive and negative pairs.

**Bimodal Loss.** The bimodal loss, denoted as $L_{\text{bimodal}}$, is defined also as the NTXent-Loss applied to the output embeddings generated by both the language model and the graph model within a given batch. This loss function aims to minimize the distance between the embeddings of the same index from both models while maximizing the distance between embeddings corresponding to different indices. To achieve this, we employ two distinct projection blocks to convert the embeddings from the graph and language models into a unified third latent space. Utilizing a single projection block to transfer the embeddings from one model into the latent space of the other could inadvertently lead to the training of one model to mimic the behavior of the other. Such an outcome is undesirable, as the distinct functionalities of the models enhance the universal applicability of the bimodal architecture.

## 5 EXPERIMENTS AND LIMITATIONS

### 5.1 PRETRAINING DATASETS AND DATA PREPARATION

We pretrain our model on parts of the PubChem (Kim et al., 2023) dataset. Initially, the compounds in them are stored in SMILES format. By splitting into chemically relevant substructures, leveraging a more physics-based input format (descriptors) and employing one of the most sophisticated language models, we achieve a significant milestone: a language model (LM) trained from scratch with 10 million entries from the PubChem

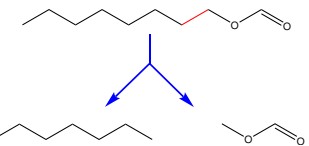

Figure 6: Example of dropping edges problem.

dataset. The pre-training process takes 98 hours for RoBERTa, about 250 hours for BERT+GCN and

Table 1: Results for classification tasks. ROC-AUC metric (higher is better) for BBBP, Tox21, ClinTox, BACE, MUV and HIV datasets, the scaffold split was used for train-test-validation (80-10-10) split.

| Models | Datasets | | | | | | |
|---|---|---|---|---|---|---|---|
| | BBBP | Tox21 (NR-AR) | ClinTox (FDA APPROVED) | ClinTox (CT TOX) | BACE | MUV | HIV |
| MolCLR(GCN) (Wang et al., 2022) | 0.723±0.025 | 0.704±0.002 | 0.668±0.035 | 0.694±0.032 | 0.711±0.090 | 0.676±0.019 | 0.787±0.005 |
| MolCLR(GIN) (Wang et al., 2022) | 0.742±0.020 | 0.740±0.003 | 0.872±0.031 | 0.775±0.037 | 0.814±0.07 | 0.796±0.017 | 0.761±0.006 |
| ChemBERTa (Chithrananda et al., 2020) | 0.647±0.053 | 0.753±0.009 | - | 0.736±0.015 | 0.721±0.022 | 0.667±0.015 | 0.625±0.012 |
| Uni-Mol (Zhou et al., 2023) | 0.729±0.006 | 0.796±0.005 | 0.895±0.018 | 0.711±0.023 | 0.857±0.002 | 0.821±0.013 | 0.808±0.003 |
| GEM (Fang et al., 2022) | 0.724±0.004 | 0.781±0.001 | 0.875±0.013 | 0.692±0.019 | 0.856±0.011 | 0.817±0.005 | 0.806±0.009 |
| GROVER (base) (Rong et al., 2020) | 0.700±0.001 | 0.743±0.001 | 0.812±0.030 | 0.664±0.032 | 0.826±0.007 | 0.673±0.018 | 0.625±0.009 |
| GROVER (large) (Rong et al., 2020) | 0.695±0.001 | 0.735±0.001 | 0.75±0.037 | 0.683±0.041 | 0.810±0.014 | 0.673±0.018 | 0.682±0.011 |
| Molformer (Wu et al., 2023) | 0.916±0.002 | - | 0.907±0.006 | 0.812±0.031 | 0.844±0.017 | - | - |
| MolFormer-XL (Ross et al., 2022) | 0.917±0.001 | 0.847±0.001 | 0.933±0.004 | 0.901±0.012 | 0.862±0.009 | - | 0.812±0.003 |
| SubD-BERT (ours) | 0.893±0.018 | 0.829±0.007 | **0.947**±0.013 | **0.926**±0.017 | 0.811±0.022 | 0.753±0.015 | 0.692±0.011 |
| BERT+GIN (ours) | **0.937**±0.002 | **0.852**±0.003 | 0.912±0.009 | 0.924±0.014 | 0.855±0.015 | **0.832**±0.011 | 0.786±0.007 |
| BERT+GCN (ours) | 0.891±0.005 | 0.830±0.002 | 0.903±0.016 | 0.793±0.031 | 0.738±0.012 | 0.794±0.017 | 0.736±0.010 |
| BERT+Graphormer (ours) | 0.862±0.009 | 0.815±0.003 | 0.878±0.019 | 0.837±0.021 | **0.892**±0.015 | 0.819±0.013 | **0.851**±0.060 |
| XGBoost (descriptors, ours) | 0.821 | 0.663 | 0.856 | 0.871 | 0.695 | 0.650 | 0.562 |
| LightGBM (descriptors, ours) | 0.832 | 0.653 | 0.886 | 0.853 | 0.682 | 0.581 | 0.546 |
| SVM (descriptors, ours) | 0.612 | 0.617 | 0.525 | 0.679 | 0.547 | 0.559 | 0.534 |

BERT+GIN, and approximately 400 hours for BERT+Graphormer (each result on the A100 GPU for 10 epochs). This model exhibits performance comparable to those trained on the largest datasets within the field. The data preparation process can be divided into two main parts.

**Language model data.** We construct the descriptors' sequence as it was described in previous paragraphs, and then we mask $15\%$ of the elements in the obtained array (after performing the tokenization process and considering them as tokens).

**Graph model data.** We build a graph based on the SMILES of the molecule and then use its augmentation, which transforms it into two different molecule graphs. The augmentation process consists of masking $20\%$ randomly chosen atom types (for GCN and GIN) and masking both $20\%$ randomly chosen atom types and edges (for Graphormer). We mask only types of atoms and edges, not the edges and atoms themselves (as in the MolCLR approach (Wang et al., 2022)), due to the greater physicochemical validity of this method. For example, if we mask the red highlighted edge in Figure 6 in octyl formate (with the SMILES encoding CCCCCCCCOC(=O)), we obtain two existing compounds: heptane (CCCCCCC) and methyl formate (COC(=O)). Thus, the model learns to converge the embeddings of octyl formate and the total embedding of heptane and methyl formate, which is fundamentally incorrect.

## 5.2 QSAR TASKS

For zero-shot evaluation, we select widely recognized cheminformatics benchmarks focused on quantitative structure-activity relationship (QSAR) tasks. Although specially designed descriptors often outperform transformer models in this context, the simplicity of these benchmarks allows for the assessment of our architecture's quality and versatility without the influence of large-scale superstructures in complex problems.

We assess four models: RoBERTa (denoted SubD-BERT), trained on descriptors; SubD-BERT combined with a Graph Isomorphism Network (GIN); SubD-BERT paired with a Graph Convolutional Network (GCN); and SubD-BERT integrated with Graphormer. All models utilize a dataset of 10 million entries from PubChem. Additionally, classical machine learning models (XGBoost, LightGBM, SVM) trained on the same descriptors provide strong baselines, known for QSAR effectiveness with handcrafted features. It can be seen that even such simple models show quite good results on many presented benchmarks, which demonstrates the correctness of the chosen paradigm of partitioning the molecule into substructures and further construction of descriptors.

The classification benchmark datasets comprised BBBP (Sakiyama et al., 2021), Tox21 (Richard et al., 2020), ClinTox (Wu et al., 2018), BACE (Wu et al., 2018), MUV (Rohrer & Baumann, 2009), and HIV (Pan et al., 2007). Performance metrics, summarized in Table 1, utilize the receiver operating characteristic area under the curve (ROC-AUC) (Table 1). The regression benchmarks included QM7 (Blum & Reymond, 2009; Rupp et al., 2012), QM8 (Ramakrishnan et al., 2015), QM9 (Ruddigkeit et al., 2012a;b), FreeSolv (Mobley & Guthrie, 2014), ESOL (Delaney, 2004), and Lipo. Evaluation metrics consisted of mean absolute error (MAE) for QM7, QM8, and QM9, and mean squared error

Table 2: Results regression tasks, MAE (less is better) metric for QM7, QM8 and QM9 datasets. MSE for FreeSolv, ESOL and Lipo, the scaffold split was used for train-test-validation (80-10-10) split.

| Models | Datasets | | | | | |
|---|---|---|---|---|---|---|
| | QM7 | QM8 (E1-CC2) | QM9 (gap) | FreeSolv | ESOL | Lipo |
| MolCLR(GCN) (Wang et al., 2022) | 85.4±2.7 | 0.0178±0.0003 | 0.0317±0.0005 | 3.259±0.261 | 1.419±0.040 | 0.957±0.010 |
| MolCLR(GIN) (Wang et al., 2022) | 91.6±3.1 | 0.0167±0.0004 | 0.0225±0.0003 | 2.884±0.249 | 1.253±0.037 | 0.651±0.004 |
| ChemBERTa (Chithrananda et al., 2020) | 177.2±4.0 | - | 0.0317±0.0005 | 3.471±0.085 | 1.487±0.107 | 0.721±0.003 |
| Uni-Mol (Zhou et al., 2023) | 41.8±0.2 | 0.0156±0.0001 | **0.0132**±0.0003 | 1.480±0.048 | 0.788±0.024 | 0.603±0.010 |
| GEM (Fang et al., 2022) | 58.9±0.8 | 0.0171±0.0001 | 0.0246±0.0003 | 1.877±0.094 | 0.798±0.029 | 0.660±0.008 |
| GROVER (base) (Rong et al., 2020) | 94.5±3.8 | 0.0218±0.0004 | 0.0197±0.0003 | 2.186±0.052 | 0.983±0.090 | 0.817±0.008 |
| GROVER (large) (Rong et al., 2020) | 92.0±0.9 | 0.0224±0.0003 | 0.0186±0.0025 | 2.272±0.051 | 0.895±0.017 | 0.823±0.001 |
| Molformer (Wu et al., 2023) | 55.2±0.8 | 0.0095±0.0005 | 0.0139±0.0004 | - | - | - |
| MolFromer-XL (Ross et al., 2022) | - | 0.0102±0.0002 | 0.0164±0.0002 | 0.571±0.027 | 0.290±0.011 | 0.551±0.002 |
| SubD-BERT (ours) | 55.4±1.3 | 0.0126±0.0003 | 0.0148±0.0003 | 0.859±0.069 | **0.292**±0.013 | **0.514**±0.003 |
| BERT+GIN (ours) | 49.8±1.1 | **0.0083**±0.0002 | 0.0145±0.0003 | **0.530**±0.041 | 0.331±0.018 | 0.526±0.009 |
| BERT+GCN (ours) | 50.1±1.7 | 0.0098±0.0003 | 0.0166±0.0004 | 0.731±0.098 | 0.357±0.026 | 0.540±0.016 |
| BERT+Graphormer (ours) | **40.6**±0.9 | 0.0114±0.0003 | 0.0134±0.0003 | 0.823±0.091 | 0.291±0.023 | 0.589±0.010 |
| XGBoost (descriptors, ours) | 69.2 | 0.0208 | 0.0174 | 5.040 | 1.233 | 1.013 |
| LightGBM (descriptors, ours) | 74.1 | 0.0203 | 0.0173 | 5.435 | 1.217 | 0.997 |
| SVR (descriptors, ours) | 143.4 | 0.0301 | 0.0334 | 6.207 | 1.830 | 1.171 |

(MSE) for FreeSolv, ESOL, and Lipo (Table 2).Three runs were conducted for each dataset, and the experiments were performed on 4 A100 GPUs.

Notably, the classification datasets predominantly involve biochemical tasks with relatively large molecules. Language models trained on SMILES representations, such as ChemBERTa, demonstrate limited efficacy in these cases, likely due to their inability to adequately capture long-range atomic interactions. In contrast, our approach, which decomposes molecules into substructures, exhibits marked improvements when supplemented with graph-based components that effectively encode structural information. Conversely, for regression tasks (in datasets such as QM8 (Ramakrishnan et al., 2015) and QM9 (Ruddigkeit et al., 2012a)) focusing on smaller molecules, language models achieve comparatively poor results individually. The number of substructures in such molecules is small, and the compounds are represented by a limited number of substructures and descriptors, which implies insufficient data. Taken together, these findings underscore the complementary contributions of both graph and language model components in optimizing predictive performance.

Graphormer, as a complex model, generally performs better on large datasets but struggles with smaller ones due to limited training data. Therefore, we recommend BERT+GIN and BERT+GCN for tasks with limited data, while BERT+Graphormer is better suited for complex tasks that require intricate node relationships.

For the general case, our models outperform other frameworks, including MolFormer-XL (Ross et al., 2022), which was trained on a 1.1 billion size dataset for approximately 3200 compute hours on Nvidia V100 GPU.

## 6 LIMITATIONS AND FURTHER IMPROVEMENTS

**Limitations.** In chemoinformatics, we frequently encounter challenges that require building predictive models from very limited training datasets, typically consisting of only 100–200 samples. In these scenarios, our models tend to be overly complex and are frequently outperformed by simpler models trained on task-specific descriptors tailored for the problem at hand.

Furthermore, similar difficulties arise in cases where the training sample size is relatively small (on the order of a few thousand samples), but the compounds under study exhibit a substantial domain shift compared to our training set, usually a subset of the PubChem database. This issue is particularly pronounced when working with inorganic compounds or polymers. Traditional pre-training approaches offer limited benefit here, as BRICS decomposition is primarily designed for organic molecules structurally similar to drug-like compounds, and its fragmentation rules require adaptation to effectively handle these chemically distinct classes.

**Future works.** Further improvements involve training the models on a larger data sample: approximately 100 million molecules or more. Given that the current approach outperforms existing methods with a rather modest training sample, this should give a significant performance gain.

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

# A    DESCRIPTORS

## A.1    GENERAL DESCRIPTION

The interplay between various molecular properties has long been recognized, and exploiting these relationships to predict molecular behavior is a standard practice in chemoinformatics (Hansch et al., 1962; 1963; Hansch & Fujita, 1964). Central to this approach is the use of molecular descriptors – quantitative representations of molecular structure and characteristics – that are typically grouped into six categories. Substructural descriptors, such as those in fingerprinting methods like MACCS keys or PubChem fingerprints, encode the presence or absence of specific structural motifs as bit vectors. Topological descriptors derive from the molecule's two-dimensional graph, capturing connectivity patterns through graph invariants. Geometric descriptors describe the three-dimensional shape and spatial configuration by considering molecular conformations or interatomic distances. Electronic descriptors relate to electron distribution and chemical reactivity, using parameters like orbital energies and atomic charges. Physico-chemical descriptors quantify bulk properties, including polarity, solubility, and hydrophobicity. Hybrid descriptors – like circular fingerprints – combine aspects of substructure and local topology to generate fixed-length numerical representations tailored to computational modeling.

Among topological descriptors, the Wiener index is particularly prominent. It measures molecular branching by summing shortest path distances between all atom pairs, originally for hydrocarbons

but now used across diverse chemical applications such as predicting boiling points and drug-likeness (Wiener, 1947). Another key descriptor is the octanol-water partition coefficient (logP), which reflects hydrophobicity through a molecule's equilibrium distribution between octanol and water phases. LogP is critical in drug design because it affects membrane permeability, solubility, and bioavailability (Estrada, 2008).

For 3D representations, the Universal Force Field (UFF) energy estimates steric strain and conformational stability by approximating atomic interactions using simplified force fields. Although less accurate than quantum mechanical calculations, UFF energy is computationally efficient for large-scale virtual screening (Rappé et al., 1992). Additionally, ring descriptors – especially distinguishing aromatic from non-aromatic rings – are vital in characterizing molecular stability, electron delocalization, and reactivity. Aromatic rings, with their conjugated $\pi$-systems, often enhance binding affinity in drug-receptor interactions, whereas non-aromatic (aliphatic) rings contribute to structural rigidity and three-dimensional shape.

These descriptors form the foundation for quantitative structure – activity and structure–property relationship (QSAR/QSPR) models, which predict molecular properties from structural data. Traditional QSAR approaches typically compute descriptors for the whole molecule, which works well for simpler compounds but may overlook subtle influences of individual substructures, especially in larger or multifunctional molecules (Estrada, 2008). To overcome this, recent strategies focus on fragment-level descriptors – generating features for discrete molecular fragments rather than the entire molecule. This fragment-based modeling enhances interpretability and allows more precise predictions by capturing localized structural effects, thereby improving our understanding of complex molecular behavior.

When employed as input representations for language models, molecular descriptors hold significant promise. Some descriptors possess positional features similar to natural language sequences while also encoding essential physicochemical properties of molecules. This dual nature allows language models to capture not only statistical patterns but also meaningful chemical relationships, thereby improving their capability to derive insights that are relevant to molecular behavior.

## A.2 SPECIFIC FOR OUR WORK

For this study, we computed a set of molecular descriptors for each substructure to serve as input features for the language model (LM). These descriptors comprehensively capture aspects of atomic composition, bonding patterns, topological indices, and physicochemical properties, providing the LM with rich, chemically meaningful representations.

**Atomic Composition**

- Counts of key atom types including Carbon (C), Hydrogen (H), Oxygen (O), Nitrogen (N), Sulfur (S), Phosphorus (P), Fluorine (F), Chlorine (Cl), Bromine (Br), Iodine (I), and Silicon (Si) – yielding 11 discrete values.

- Aggregate count of metal atoms such as Sodium (Na), Potassium (K), Magnesium (Mg), Iron (Fe), and Zinc (Zn) combined into a single value.

**Substructure Size**

- Total atom count within the substructure (single value).

**Bonding Patterns**

- Counts of various bond types: single, double, triple, and aromatic, represented as four discrete values.

- Total number of bonds in the substructure (single value).

**Topological Descriptors**

- **Wiener index**: Computed as the sum of shortest topological distances between all pairs of atoms within the substructure, serving as a quantitative measure of molecular branching Wiener (1947).

**Physicochemical Properties**

- **logP** (octanol-water partition coefficient): Classified into seven discrete bins reflecting hydrophobicity levels, as detailed in Table 3. This categorization facilitates interpretability and relevance to biological contexts.

| Category | logP Range | Description |
|---|---|---|
| 1 | $< -2.0$ | Extreme hydrophilic |
| 2 | $-2.0 \leq \text{logP} < -0.5$ | Strong hydrophilic |
| 3 | $-0.5 \leq \text{logP} < 0.0$ | Moderate hydrophilic |
| 4 | $0.0 \leq \text{logP} < 1.0$ | Neutral |
| 5 | $1.0 \leq \text{logP} < 2.0$ | Moderate hydrophobic |
| 6 | $2.0 \leq \text{logP} < 4.0$ | Strong hydrophobic |
| 7 | $\geq 4.0$ | Extreme hydrophobic |

Table 3: logP categorization scheme

- **Universal Force Field (UFF) energy**: Categorized into seven discrete ranges representing steric strain and conformational stability; see Table 4. Despite its approximate nature compared to quantum calculations, UFF energy offers a computationally efficient descriptor of molecular geometry.

| Category | Energy Range (kcal/mol) | Description |
|---|---|---|
| 1 | $< 0$ | Stable complex |
| 2 | $0 \leq E < 50$ | Very stable |
| 3 | $50 \leq E < 100$ | Stable |
| 4 | $100 \leq E < 150$ | Moderate |
| 5 | $150 \leq E < 200$ | Unstable |
| 6 | $200 \leq E < 300$ | High energy |
| 7 | $\geq 300$ | Extreme energy |

Table 4: UFF energy categorization scheme

**Ring Systems**

- Number of atoms that participate in at least one ring (single value).
- Number of complete rings fully contained within the substructure (single value).

In summary, this results in a feature vector of dimension 23 for each molecular substructure, integrating diverse structural and physicochemical information suitable for language model input.

## B BRICS DECOMPOSITION METHODOLOGY

The Breaking of Retrosynthetically Interesting Chemical Substructures (BRICS) method (Degen et al., 2008) offers a systematic, rule-based framework for fragmenting molecules into chemically meaningful components that correspond to synthetic building blocks. This fragmentation strategy is founded on the principles of retrosynthetic analysis (Corey, 1991), wherein complex molecules are iteratively broken down into simpler precursors by cleaving bonds commonly formed or targeted in synthetic reactions.

### B.1 BRICS CLEAVAGE RULES

The BRICS algorithm applies a set of 16 predefined bond cleavage rules designed to target specific chemical environments. These rules selectively break bonds adjacent to key atoms, ensuring the

preservation of functional group integrity and chemical relevance. The full repertoire of bonds eligible for cleavage includes:

- Single bonds between aliphatic carbon atoms and any of the following heteroatoms:
    - C–N bonds (Rule 1)
    - C–O bonds (Rule 2)
    - C–S bonds (Rule 3)
    - C–P bonds (Rule 4)
- Bonds in cyclic systems:
    - C=C in conjugated systems (Rule 5)
    - Aromatic C–N bonds (Rule 6)
    - Aromatic C–O bonds (Rule 7)
    - Aromatic C–S bonds (Rule 8)
- Bonds adjacent to carbonyl groups:
    - C–C(=O) (amide bonds, Rule 9)
    - N–C(=O) (peptide bonds, Rule 10)
    - O–C(=O) (ester bonds, Rule 11)
- Specialized cleavages:
    - C≡C triple bonds (Rule 12)
    - C–Si bonds (Rule 13)
    - S(=O)–N sulfonamide bonds (Rule 14)
    - C–B boronic ester bonds (Rule 15)
    - C–Sn stannane bonds (Rule 16)

Each cleavage generates molecular fragments that incorporate dummy atoms at the original attachment sites, thereby retaining critical information about potential points for recombination. These cleavage rules are applied recursively until no additional bonds meeting the criteria remain, resulting in a comprehensive set of BRICS fragments.

## B.2 CHEMICAL RATIONALE

The choice of targeted bond types is grounded in their widespread occurrence in synthetic organic chemistry (Carey & Sundberg, 2007) and pharmaceutical molecules (Kumar et al., 2012). This targeted fragmentation approach offers several key advantages, including:

- Preservation of essential pharmacophoric features within the generated fragments, ensuring retention of bioactive characteristics.
- Production of building blocks that are synthetically accessible, facilitating practical chemical synthesis.
- Maintenance of chemically valid valency states in all fragments, preserving structural integrity and realism.
- Compatibility with combinatorial library design workflows, enabling efficient exploration of chemical space (Kodadek, 2011).

## B.3 APPLICATION IN DESCRIPTOR CALCULATION

In our preprocessing pipeline, molecules are first decomposed into substructures using the BRICS methodology, after which molecular descriptors are computed for each fragment individually. This fragment-based approach provides several distinct analytical advantages:

- Enables more precise characterization of local chemical environments that predominantly influence specific molecular properties.
- Facilitates enhanced interpretability by allowing property attributions at the fragment level.

Table 5: Results of comparison of models trained on molecular representations with (highlighted with green color) and without (without highlighting) additional substructures for regression tasks, MAE (less is better) metric for QM7, QM8 and QM9 datasets. MSE for FreeSolv, ESOL and Lipo, the scaffold split was used for train-test-validation (80-10-10) split.

| Models | Datasets | | | | | |
|---|---|---|---|---|---|---|
| | QM7 | QM8 (E1-CC2) | QM9 (gap) | FreeSolv | ESOL | Lipo |
| SubD-BERT (with additional) | 55.4±1.3 | 0.0126±0.0003 | 0.0148±0.0003 | 0.859±0.069 | **0.292**±0.013 | **0.514**±0.003 |
| BERT+GIN (with additional) | 49.8±1.1 | **0.0083**±0.0002 | 0.0145±0.0003 | **0.530**±0.041 | 0.331±0.018 | 0.526±0.009 |
| BERT+GCN (with additional) | 50.1±1.7 | 0.0098±0.0003 | 0.0166±0.0004 | 0.731±0.098 | 0.357±0.026 | 0.540±0.016 |
| BERT+Graphormer (with additional) | 40.6±0.9 | 0.0114±0.0003 | 0.0134±0.0003 | 0.823±0.091 | 0.291±0.023 | 0.589±0.010 |
| XGBoost (descriptors, with additional) | 69.2 | 0.0208 | 0.0174 | 5.040 | 1.233 | 1.013 |
| LightGBM (descriptors, with additional) | 74.1 | 0.0203 | 0.0173 | 5.435 | 1.217 | 0.997 |
| SVR (descriptors, with additional) | 143.4 | 0.0301 | 0.0334 | 6.207 | 1.830 | 1.171 |
| SubD-BERT (without additional) | 53.1±1.2 | 0.0128±0.0002 | 0.0153±0.0004 | 0.881±0.072 | 0.297±0.011 | 0.525±0.006 |
| BERT+GIN (without additional) | 47.9±0.9 | 0.0087±0.0003 | 0.0148±0.0004 | 0.569±0.039 | 0.356±0.017 | 0.529±0.012 |
| BERT+GCN (without additional) | 49.7±1.4 | 0.0097±0.0002 | 0.0172±0.0006 | 0.788±0.091 | 0.371±0.029 | 0.543±0.017 |
| BERT+Graphormer (without additional) | **39.6**±0.9 | 0.0112±0.0003 | **0.0132**±0.0003 | 0.830±0.087 | 0.313±0.020 | 0.594±0.013 |
| XGBoost (descriptors, without additional) | 68.1 | 0.0210 | 0.0175 | 5.971 | 1.439 | 1.075 |
| LightGBM (descriptors, without additional) | 72.1 | 0.0204 | 0.0175 | 6.304 | 1.292 | 1.051 |
| SVR (descriptors, without additional) | 136.9 | 0.0289 | 0.0343 | 8.049 | 3.088 | 1.319 |

Table 6: Results of comparison of models trained on molecular representations with (highlighted with green color) and without (without highlighting) additional substructures for classification tasks. ROC-AUC metric (higher is better) for BBBP, Tox21, ClinTox, BACE, MUV and HIV datasets, the scaffold split was used for train-test-validation (80-10-10) split.

| Models | Datasets | | | | | | |
|---|---|---|---|---|---|---|---|
| | BBBP | Tox21 (NR-AR) | ClinTox (FDA APPROVED) | ClinTox (CT TOX) | BACE | MUV | HIV |
| SubD-BERT (with additional) | 0.893±0.018 | 0.829±0.007 | **0.947**±0.013 | **0.926**±0.017 | 0.811±0.022 | 0.753±0.015 | 0.692±0.011 |
| BERT+GIN (with additional) | 0.937±0.002 | **0.852**±0.003 | 0.912±0.009 | **0.924**±0.014 | 0.855±0.015 | **0.832**±0.011 | 0.786±0.007 |
| BERT+GCN (with additional) | 0.891±0.005 | 0.830±0.002 | 0.903±0.016 | 0.793±0.031 | 0.738±0.012 | 0.794±0.017 | 0.736±0.010 |
| BERT+Graphormer (with additional) | 0.862±0.009 | 0.815±0.003 | 0.878±0.019 | 0.837±0.021 | **0.892**±0.015 | 0.819±0.013 | **0.851**±0.060 |
| XGBoost (descriptors, with additional) | 0.821 | 0.663 | 0.856 | 0.871 | 0.695 | 0.650 | 0.562 |
| LightGBM (descriptors, with additional) | 0.832 | 0.653 | 0.886 | 0.853 | 0.682 | 0.581 | 0.546 |
| SVM (descriptors, with additional) | 0.612 | 0.617 | 0.525 | 0.679 | 0.547 | 0.559 | 0.534 |
| SubD-BERT (without additional) | 0.871±0.024 | 0.821±0.005 | 0.893±0.007 | 0.901±0.012 | 0.803±0.016 | 0.738±0.008 | 0.678±0.014 |
| BERT+GIN (without additional) | **0.941**±0.004 | 0.842±0.007 | 0.899±0.011 | 0.911±0.008 | 0.847±0.006 | 0.819±0.014 | 0.761±0.004 |
| BERT+GCN (without additional) | 0.897±0.009 | 0.811±0.003 | 0.873±0.012 | 0.787±0.026 | 0.789±0.007 | 0.797±0.012 | 0.729±0.008 |
| BERT+Graphormer (without additional) | 0.882±0.012 | 0.831±0.009 | 0.851±0.006 | 0.816±0.018 | 0.871±0.017 | 0.822±0.011 | 0.807±0.048 |
| XGBoost (descriptors, without additional) | 0.847 | 0.652 | 0.788 | 0.803 | 0.691 | 0.637 | 0.526 |
| LightGBM (descriptors, without additional) | 0.824 | 0.649 | 0.810 | 0.804 | 0.699 | 0.542 | 0.526 |
| SVM (descriptors, without additional) | 0.548 | 0.561 | 0.518 | 0.526 | 0.596 | 0.526 | 0.561 |

- Improves the treatment of structurally complex molecules, where different subregions may exert contrasting effects on target properties.

- Aligns naturally with fragment-based drug design strategies, promoting integration with established pharmaceutical workflows (Bian & Xie, 2018).

The fragment-centric representation is consistent with the concept of molecular signatures, in which molecular properties arise from both additive and nonlinear interactions among constituent substructures. By individually evaluating these fragments, our model is capable of pinpointing key structural motifs that significantly impact the target properties, while simultaneously ensuring synthetic feasibility through the use of the BRICS framework.

## C  ADDITIONAL EXPERIMENTAL RESULTS

We compared models trained on molecular representations with and without additional substructures; Tables 5 and 6 summarize the results for both settings. Overall, including extra substructures typically produces modest improvements, but this benefit is not universal. For some datasets – particularly those composed of relatively small molecules – the additional overlapping substructures do not meaningfully enhance connectivity and therefore provide little or no advantage. Consequently, the effectiveness of adding substructures depends on molecule size and on how much improved substructure connectivity contributes to the prediction task.

# D GRAPH MODELS

**GCN.** The Graph Convolutional Network (GCN), as introduced by Kipf and Welling (Kipf & Welling, 2016), constitutes a significant advancement in the field of graph neural networks, employing convolutional operations tailored specifically for graph data structures. Distinct from conventional neural networks that utilize linear transformations through a weight matrix $\mathbf{W}$, represented mathematically as $h = \mathbf{W}x$, GCNs incorporate the inherent topological characteristics of the graph to update node representations. This approach is particularly advantageous given the phenomenon of network homophily, wherein connected nodes are more likely to exhibit similar attributes.

GCNs operate through a principle known as neighborhood aggregation, which amalgamates the features of a target node with those of its neighboring nodes. For a given node $i$ and its associated neighborhood $N_i$, this aggregation is formalized as follows:

$$h_i = \sum_{j \in N_i} \mathbf{W}x_j. \tag{3}$$

This formulation enables GCNs to enhance the feature representation of each node by leveraging the attributes of its direct connections. However, given the variability in node degree, it is essential to normalize the aggregated features to ensure comparability across nodes. This normalization is achieved by factoring in the degree of the node, leading to the expression:

$$h_i = \frac{1}{\deg(i)} \sum_{j \in N_i} \mathbf{W}x_j. \tag{4}$$

Kipf et al. further refined the GCN architecture by addressing the potential imbalance in feature propagation, whereby nodes with a greater number of neighbors may disproportionately influence the learning process. To mitigate this effect, they proposed a weighted aggregation mechanism that accounts for the degrees of both the target node and its neighbors. The updated formulation is expressed as:

$$h_i = \sum_{j \in N_i} \frac{1}{\sqrt{\deg(i)\deg(j)}} \mathbf{W}x_j. \tag{5}$$

This enhancement promotes a more equitable distribution of influence among nodes, thereby ensuring that features from less-connected nodes are adequately considered.

The versatility of GCNs has led to their incorporation in various advanced frameworks, including Graph Attention Networks (GAT) (Velickovic et al., 2017) and Message Passing Neural Networks (MPNN). Their capacity to capture complex relational patterns and dependencies within graph structures renders GCNs particularly suited for applications spanning diverse domains, such as social network analysis, recommendation systems, and molecular property prediction in cheminformatics.

Additionally, GCNs can be further refined through modifications such as attention mechanisms that differentially weight the contributions of neighboring nodes based on learned significance or by integrating diverse edge types to enrich the contextual information. These adaptations contribute to the ongoing research aimed at enhancing GCN performance across a wide spectrum of graph-related tasks. In the context of our model, GCNs are instrumental in leveraging the structural information inherent in molecular graphs, facilitating improved predictive accuracy with respect to compound properties.

**Graph Isomorphism Network (GIN).** The Graph Isomorphism Network (GIN) is a neural network architecture introduced by Xu et al (Xu et al., 2018). in 2019 that aims to improve the expressive capabilities of graph neural networks (GNNs). GIN is particularly significant due to its equivalence to the Weisfeiler-Lehman (WL) graph isomorphism test, which serves as a standard for assessing the ability of models to distinguish between different graph structures.

The update mechanism for GIN aggregates node features and those of their neighbors using the following formulation:

$$h_v^{(k)} = \text{MLP}^{(k)} \left( (1 + \varepsilon)h_v^{(k-1)} + \sum_{u \in \mathcal{N}(v)} h_u^{(k-1)} \right) \tag{6}$$

In this equation, $h_v^{(k)}$ denotes the representation of node $v$ at the $k$-th layer, while $\mathcal{N}(v)$ represents the set of neighboring nodes. The term $\text{MLP}^{(k)}$ indicates a multi-layer perceptron applied to the aggregated features. The parameter $\epsilon$ is incorporated to preserve the unique identity of node features, thereby enhancing the model's ability to differentiate between nodes based on their characteristics.

GIN operates using a two-step framework: initially performing aggregation of neighboring features, followed by the application of a multi-layer perceptron. This approach facilitates the learning of complex representations that capture both local and relational information within graph structures.

Empirical evaluations of GIN demonstrate its superior performance in graph classification tasks compared to other GNN variants, underscoring its robustness across various datasets. The architecture coalesces well with applications where fine distinctions in graph structures are essential, such as in the prediction of molecular properties.

In this study, the integration of GIN into our model is anticipated to enhance the ability to capture intricate relationships within molecular graphs. This choice aims to improve the predictive performance across diverse physicochemical tasks, contributing to a more accurate assessment of chemical compounds.

**Graphormer.** Graphormer is an advanced architecture designed to enhance the capabilities of the Transformer model specifically for graph representation learning, as introduced by Ying et al. (Ying et al., 2021) This architecture effectively addresses the limitations encountered by traditional Transformer models, which often struggle to capture the inherent structural information present in graph data. To this end, Graphormer incorporates several innovative mechanisms, including centrality encoding, spatial encoding, and edge encoding, thereby improving the representation of graph data.

1. Centrality Encoding: Graphormer enhances the feature representation of nodes by integrating degree centrality into the input features. For a node $v$, the encoded feature is defined as:

$$h_v^{\text{centrality}} = h_v + \text{MLP}(\deg(v)), \tag{7}$$

where $h_v$ represents the original feature vector of node $v$, $\deg(v)$ denotes the degree of node $v$, and MLP denotes a multi-layer perceptron that transforms the centrality information into a vector space that aligns with the node features.

2. Spatial Encoding: The architecture utilizes spatial encoding to represent the shortest path distance (SPD) between nodes. The SPD between nodes $u$ and $v$ is computed and expressed as:

$$\text{spatial}(u, v) = \frac{1}{\text{SPD}(u, v) + 1}, \tag{8}$$

where $\text{SPD}(u, v)$ denotes the shortest path distance between nodes $u$ and $v$.

3. Edge Encoding: To effectively utilize the significance of edge features, Graphormer incorporates edge encoding by calculating the interaction between edge features and node embeddings. This edge encoding is defined as:

$$e(u, v) = \frac{\text{dot}(h_u \cdot W_Q, h_v \cdot W_K)}{\sqrt{d}}, \tag{9}$$

where $e(u, v)$ represents the embedded feature for the edge connecting nodes $u$ and $v$, $W_Q$ and $W_K$ are query and key martices respectively, d corresponds to the hidden dimension. This interaction is integrated into the attention mechanism by modifying the attention score as follows:

$$\text{Attention}(u, v) = \frac{\exp(e(u, v) + \text{spatial}(u, v))}{\sum_{w \in \mathcal{N}(u)} \exp(e(u, w) + \text{spatial}(u, w))} \cdot V, \tag{10}$$

where $\mathcal{N}(u)$ represents the set of neighbors of node $u$ and $V$ is value matrix.

Graphormer has exhibited state-of-the-art performance across a variety of graph-level tasks, including graph classification and molecular property prediction, demonstrating its versatility and robustness. By integrating Graphormer into our model, we leverage its advanced mechanisms to accurately capture intricate relationships and patterns within molecular graphs, significantly enhancing predictive performance across a broad spectrum of physicochemical tasks.

## E  SOME TRAINING DETAILS

**Weighted Cross-Entropy Loss.**    Weighted cross-entropy loss assigns different weights to different classes based on their frequency in the dataset. Such approach is useful when you have unbalanced data and you want the model to pay more attention to less represented classes. Class weights do compensate for the imbalance by increasing the contribution of rare classes to the total loss, according to the formulae:

$$L = -\frac{1}{N} \sum_{i=1}^{N} \sum_{c=1}^{C} w_c \cdot y_{i,c} \cdot \log(p_{i,c} + \epsilon), \tag{11}$$

where
- $N$ - the number of examples in the batches,
- $C$ - number of classes,
- $w_c$ - weight for class $c$,
- $y_{i,c}$ - true label for example $i$ and class $c$,
- $p_{i,c}$ - probability predicted by the model for example $i$ and class $c$ (after applying softmax),
- $\epsilon$ - a small value to prevent division by zero.

This formulae calculates the average of the weighted cross-entropy over all examples in the batches. We used this variation of Cross-Entropy Loss for the HIV, the Tox21, the ClinTox and the MUV datasets to improve the quality of our models.

## F  TESTING DATASETS (QSAR)

**QM7.**    The QM7 dataset is a curated subset of GDB-13, a comprehensive database containing nearly one billion stable and synthetically accessible organic molecules. Specifically, QM7 includes 7,165 molecules, each composed of up to 23 atoms, with a focus on seven heavy atoms: carbon (C), nitrogen (N), oxygen (O), and sulfur (S). This dataset not only provides a diverse array of molecular structures – such as double and triple bonds, cyclic compounds, carboxylic acids, cyanides, amides, alcohols, and epoxides – but also features the Coulomb matrix representation of these molecules. Additionally, the atomization energies for the QM7 molecules are computed using methods aligned with the FHI-AIMS implementation of the Perdew-Burke-Ernzerhof hybrid functional (PBE0).

**QM8.**    The QM8 dataset consists of 21,786 small organic molecules and serves as a critical resource for evaluating machine learning models in predicting quantum mechanical properties. Each molecule is characterized by quantum chemical properties, including total energies and electronic spectra derived from time-dependent density functional theory (TDDFT). Although TDDFT offers favorable computational efficiency for predicting electronic spectra across chemical space, its accuracy can be limited.dataset is used to validate machine learning models in a prediction of deviations between TDDFT predictions and reference second-order approximate coupled-cluster (CC2) singles and doubles spectra. This approach has successfully applied to the low-lying singlet-singlet vertical electronic spectra of over 20,000 synthetically feasible small organic molecules.

**QM9.**    The QM9 dataset is a prominent collection in computational chemistry, comprising 133,885 molecules with up to nine heavy atoms, including carbon (C), nitrogen (N), oxygen (O), and fluorine (F). This dataset is particularly valuable for evaluating machine learning models as it features a rich set of molecular structures representative of a wide chemical space.

Each molecule is identified by a unique 'gdb9' tag facilitating data extraction and a consecutive integer identifier (i). Rotational constants (A, B, and C, in GHz) describe the molecule's rotational inertia. The dipole moment ($\mu$, in Debye) indicates the molecule's polarity, while isotropic polarizability ($\alpha$, in $a^3$) reflects its response to electric fields. The energies of the highest occupied molecular orbital (HOMO) and lowest unoccupied molecular orbital (LUMO), both in Hartree (Ha), are included, along with the energy gap ($lumo - homo$, also in Ha). Electronic spatial extent ($R^2$, in Ha) characterizes the molecule's size. Vibrational properties are represented by the zero-point vibrational energy ($zpve$, in Ha). Thermodynamic properties at 0 K and 298.15 K are also provided, including internal energy ($U_0$ and $U$, in Ha), enthalpy ($H$, in Ha), Gibbs free energy (G, in Ha), and heat capacity ($Cv$, in cal/mol K).

**FreeSolv.** The FreeSolv database is a comprehensive resource that offers a curated collection of experimental and calculated hydration-free energies for small neutral molecules in water. This database integrates both experimental values obtained from established literature and calculated values derived from advanced molecular dynamics simulations. It encompasses 643 small molecules, significantly expanding upon a previously existing dataset of 504 molecules. FreeSolv includes essential metadata, such as molecular structures, input files, and annotations, facilitating ease of access and reproducibility in research. The calculated values are derived from alchemical free energy calculations employing the Generalized Amber Force Field (GAFF) within a TIP3P water model, utilizing AM1-BCC charges. Calculations were conducted using the GROMACS simulation package, ensuring high accuracy and reliability. Furthermore, the database is regularly updated with new experimental references and data, enhancing its utility as a dynamic and evolving resource for the research community. Detailed construction processes and references are documented to provide transparency and context for users.

**ESOL.** The ESOL (Estimated SOLubility) dataset, introduced by Delaney ((Delaney, 2004)), provides a robust method for estimating the aqueous solubility of compounds directly from their molecular structure. The model, derived from a comprehensive training set of 2,874 measured solubilities, employs linear regression analysis based on nine molecular properties, with calculated logP octanol identified as the most significant parameter. Other key descriptors include molecular weight, the proportion of heavy atoms in aromatic systems, and the number of rotatable bonds. ESOL demonstrates competitive performance relative to the well-established General Solubility Equation, particularly for medicinal and agrochemical compounds. In our study, we build upon the ESOL dataset by utilizing a superstructure aimed at predicting water solubility across an extended set of 1,128 samples. This enhancement not only broadens the applicability of the original model but also supports more precise solubility estimations in diverse chemical spaces. The combination of ESOL's foundational framework with our superstructure facilitates further exploration of solubility-related properties, making it a valuable tool for researchers in drug discovery and environmental sciences.

**LIPO (Lipophilicity).** The lipophilicity dataset is a vital resource for examining the pharmacokinetic properties of drug molecules, specifically in relation to membrane permeability and solubility. Curated from the ChEMBL database, this dataset encompasses experimental results for the octanol/water distribution coefficient (logD) at pH 7.4 across a diverse collection of 4,200 compounds. Lipophilicity, described by the n-octanol/water partition coefficient or the n-octanol/buffer solution distribution coefficient, is of considerable significance in pharmacology, toxicology, and medicinal chemistry. In this study, a quantitative structure–property relationship (QSPR) analysis was conducted to predict logD values at pH 7.4 for the dataset. Comparative analysis with previously established logD values demonstrated that the developed predictive model offers reliable and robust performance. This enhances its utility as a valuable tool for researchers aiming to evaluate and optimize the lipophilicity of potential drug candidates, thereby informing pharmacological strategies in drug development.

**BBBP.** The Blood-Brain Barrier Permeability (BBBP) dataset serves as a resource for studying the ability of chemical compounds to penetrate the blood-brain barrier (BBB), which is an important consideration in drug development for central nervous system disorders. The BBB selectively regulates the transfer of substances from the bloodstream into the brain, thereby necessitating an accurate assessment of BBB penetration for potential therapeutic agents. In this study, the original BBBP dataset was modified to create both free-form and in-blood-form datasets. Molecular descriptors were generated for each dataset and employed in machine learning (ML) models to predict BBB penetration. The dataset was partitioned into training, validation, and test sets using the scaffold split algorithm from MoleculeNet, which intentionally creates an unbalanced partition to enhance the evaluation of predictive performance for compounds that are structurally dissimilar to those used in the training data. Notably, the random forest model achieved the highest prediction score using 212 descriptors from the free-form dataset, surpassing previous benchmarks derived from the same splitting method without any external database augmentations. Additionally, a deep neural network produced comparable results with just 11 descriptors, emphasizing the significance of recognizing glucose-like characteristics in the prediction of BBB permeability.

**Tox21.** The Tox21 dataset is a significant resource in toxicology research, comprising 12,060 training samples and 647 test samples representing various chemical compounds. Each sample is associated with 12 binary labels reflecting the outcomes (active/inactive) of different toxicological experiments, although the label matrix contains numerous missing values. Due to the extensive size of the dataset, our study focuses exclusively on predicting the NR-AR property. Since its inception in 2009, the Tox21 project has screened approximately 8,500 chemicals across more than 70 high-throughput assays, yielding over 100 million data points, all publicly accessible through partner organizations such as the United States Environmental Protection Agency (EPA), National Center for Advancing Translational Sciences (NCATS), and National Toxicology Program (NTP). This collaborative effort has produced the largest compound library specifically aimed at enhancing understanding of the chemical basis of toxicity across research and regulatory domains. Each federal partner contributed specialized resources, culminating in a diverse set of compound libraries that collectively expand coverage of chemical structures, use categories, and properties. The integrated approach of Tox21 enables comprehensive analysis of structure–activity relationships through ToxPrint chemotypes, allowing the identification of activity patterns that might otherwise remain undetected. This dataset underscores the central premise of the Tox21 program: that collaborative merging of distinct compound libraries yields greater insights than could be achieved in isolation.

**ClinTox.** The ClinTox dataset serves as an a resource for understanding the factors influencing drug approval and toxicity outcomes in clinical trials. This dataset compares drugs approved by the FDA with those that have failed clinical trials due to toxicity reasons, encompassing two classification tasks for 1,491 drug compounds with known chemical structures. Specifically, it aims to classify (1) clinical trial toxicity (or absence of toxicity) and (2) FDA approval status. The compilation of FDA-approved drugs is derived from the SWEETLEAD database, while information regarding compounds that failed clinical trials is sourced from the Aggregate Analysis of Clinical Trials (AACT) database.

**BACE.** The BACE dataset is a resource for the study of inhibitors targeting human $\beta$-secretase 1 (BACE-1), a key enzyme involved in the pathogenesis of Alzheimer's disease. This dataset provides both quantitative binding results (IC50 values) and qualitative outcomes (binary labels) for a collection of 1,522 compounds, encompassing experimental values reported in the scientific literature over the past decade. Notably, some of these compounds have detailed crystal structures available, which enhances the dataset's utility for structure-activity relationship (SAR) studies. The BACE dataset has been integrated into MoleculeNet, where it is structured as a classification task, effectively merging the compounds with their corresponding 2D structures and binary labels. The use of scaffold splitting in this context is particularly beneficial, facilitating the assessment of predictive performance on a single protein target by preventing bias associated with structural similarities among compounds. This integration of experimental binding data and diverse structural information underscores the dataset's potential to aid in the design and optimization of BACE-1 inhibitors, ultimately contributing to advancements in therapeutic strategies for Alzheimer's disease.

**MUV.** The Maximum Unbiased Validation (MUV) dataset serves as a benchmark for evaluating virtual screening techniques in drug discovery. Selected from the PubChem BioAssay database, the MUV dataset comprises 17 challenging tasks associated with approximately 90,000 chemical compounds, strategically designed to facilitate robust validation of virtual screening methodologies. A key feature of this dataset is its foundation in refined nearest neighbor analysis, a technique derived from spatial statistics that offers a mathematical framework for the nonparametric analysis of mapped point patterns. This methodology enables the systematic design of benchmark datasets by purging compounds that exhibit activity against pharmaceutically relevant targets while eliminating unselective hits. Through topological optimization and experimental design strategies, the refined nearest neighbor analysis constructs data sets of active compounds and decoys, ensuring they are unbiased concerning analogue bias and artificial enrichment. Consequently, the MUV dataset provides an essential resource for Maximum Unbiased Validation, empowering researchers to assess and improve the predictive performance of virtual screening methods in a more rigorous manner.

**HIV.** The HIV dataset, introduced by the Drug Therapeutics Program (DTP) AIDS Antiviral Screen, encompasses an extensive screening of over 40,000 compounds to assess their inhibitory effects on HIV replication. The screening results are categorized into three classifications: confirmed inactive (CI), confirmed active (CA), and confirmed moderately active (CM). For the purposes of

analysis, CA and CM labels are combined to formulate a binary classification task distinguishing between inactive (CI) and active (CA/CM) compounds. This dataset is particularly valuable for researchers aiming to discover new categories of HIV inhibitors, and the use of scaffold splitting is recommended to enhance the identification of novel compounds while mitigating bias related to structural similarities. Additionally, the HIV positive selection mutation database provides a comprehensive resource for understanding the selection pressures exerted on HIV protease and reverse transcriptase, which are critical targets for antiretroviral therapy. This large-scale database contains sequences from approximately 50,000 clinical AIDS samples, leveraging contributions from Specialty Laboratories, Inc., allowing for high-resolution selection pressure mapping. It offers insights into selection pressures at individual sites and their interdependencies, along with datasets from other public repositories, such as the Stanford HIV database. This confluence of data facilitates cross-validation with independent datasets and enables a nuanced evaluation of drug treatment effects, significantly advancing the understanding of HIV resistance mechanisms.

## G CODE

Our code is available by link [1].

## H LLM USAGE

We used LLMs for editing grammatical errors in the text of this paper.

---

[1]Our code for all experiments is accessible on https://anonymous.4open.science/r/thinking-like-a-chemist-EC7B.

