# OpenReview forum: "Thinking like a CHEMIST: Combined Heterogeneous Embedding Model Integrating Structure and Tokens"
_ICLR.cc/2026/Conference — ICLR 2026 Conference Withdrawn Submission_

### Official Review · Reviewer_Ye9W · 2025-10-30

**Soundness:** 2
**Presentation:** 2
**Contribution:** 2
**Rating:** 2
**Confidence:** 4

**Summary:**

This paper introduces a bimodal framework for molecular representation learning that integrates structure-based and descriptor-based approaches. The authors propose decomposing molecules into chemically meaningful substructures using the BRICS method, computing physicochemical descriptors for each fragment, and using these as input to a language model. Simultaneously, graph neural networks (GIN, GCN, or Graphormer) process molecular graphs, with both modalities aligned via a contrastive learning objective. The model is pre-trained on PubChem and evaluated on QSAR benchmarks.

**Strengths:**

This paper is well written and easy to understand.

**Weaknesses:**

1. ​​Insufficient Discussion of Related Work:​​ The paper does not adequately situate itself within the growing field of ​​fragment- or motif-based molecular pre-training​​. There is no discussion or comparison with models like FineMolTex [1], MoleculeSTM [2], and MolCA [3], which have a very similar bimodal design. This omission weakens the claim of novelty.

2. Narrow Scope of Evaluation:​​ The experimental validation is limited to standard property prediction tasks (QSAR). For a bimodal model, its potential is not fully tested on ​​cross-modal tasks​​ that could truly showcase its integrated understanding, such as text-based molecule generation or molecule captioning. The absence of such benchmarks leaves the "bimodal" capability somewhat underdemonstrated.

3. ​​Incomplete and Somewhat Outdated Baselines:​​ The set of baseline models is not comprehensive. It lacks comparisons with recent and highly relevant ​​multimodal molecular models​​ . Benchmarking against these stronger contemporaries is crucial to claim state-of-the-art performance convincingly.

[1] Advancing Molecular Graph-Text Pre-training via Fine-grained Alignment

[2] Multi-modal Molecule Structure-text Model for Text-based Retrieval and Editing

[3] MolCA: Molecular Graph-Language Modeling with Cross-Modal Projector and Uni-Modal Adapter

**Questions:**

1. Given the existence of numerous pre-trained Graph Neural Network (GNN) models for molecules such as [1], what is the specific rationale for the authors to conduct a separate pre-training phase for the GNN component in their work?

2. The proposed bimodal loss function is a critical component of your framework. While you mention that α, β, and γ are hyperparameters with default values of 1.0, the paper does not provide a detailed analysis of their impact. Could you please elaborate on the following aspects?

[1] Pre-training Molecular Graph Representation with 3D Geometry

---

### Official Review · Reviewer_YtmJ · 2025-10-31

**Soundness:** 3
**Presentation:** 3
**Contribution:** 2
**Rating:** 4
**Confidence:** 3

**Summary:**

This paper proposes a novel approach to molecular representation learning by reframing molecules as sequences of chemically meaningful substructures. The core idea is to decompose molecules using the BRICS algorithm, calculate a set of 23 physicochemical and topological descriptors for each substructure, and use these descriptor sequences as input to a language model (RoBERTa). This "SubD-BERT" model forms the foundation, which is then extended into several bimodal architectures by combining it with graph neural networks (GCN, GIN, Graphormer). The graph models are trained with a combination of feature masking and contrastive learning. The two modalities are fused using projection blocks and a bimodal contrastive loss.

**Strengths:**

1. The idea of using BRICS fragmentation to create a "chemical vocabulary" and then describing each "word" (substructure) with a rich set of descriptors is creative and well-motivated. The argument for aligning the model's "thinking" with a chemist's fragment-based reasoning is compelling.

2.The paper is generally well-written and clear. The figures effectively illustrate the overall architecture and key processes like tokenization and graph augmentation. The methodology is explained in a logical, step-by-step manner that should be reproducible for researchers in the field.

3.The paper makes a strong case that its descriptor-based approach overcomes key limitations of SMILES, such as the inability to represent certain compounds (e.g., metal complexes) and the loss of physicochemical semantics.

**Weaknesses:**

1. A significant weakness is the lack of discussion and comparison with other recent multi-modal molecular models. The related work section and experiments focus on unimodal (SMILES-based LMs or GNNs) and simpler bimodal (SMILES+Graph) models. However, several advanced multi-modal frameworks have been proposed that also aim to fuse different molecular perspectives. Notably:

a. MoleculeSTM (Liu et al., Nature Machine Intelligence 2023) is a multi-modal model that aligns molecular structures with natural language text in a shared embedding space. While its modality (text) is different, its methodology for cross-modal alignment is a relevant point of discussion for any work on bimodal learning in chemistry.

b. 3DToMolo (Zhang et al., BMC Bioinformatics 2025) explicitly incorporates 3D structural information with text descriptions, focusing on a flexible, substructure-aware framework. The emphasis on substructures makes it a particularly relevant contemporary to compare and contrast with the philosophy of this work.

The omission of a discussion on how this work relates to or advances beyond such multi-modal paradigms is a notable gap.

2.The custom tokenization scheme for the descriptor sequences is a critical component, but its justification is somewhat heuristic. The paper states that standard tokenization like BPE is unnecessary, but it doesn't provide an ablation or empirical evidence showing that this custom method is superior to a more standard approach (e.g., treating each floating-point descriptor as a token after normalization/discretization, or using a simple linear layer). A comparison would strengthen the claim that this specific tokenization is optimal.

3.The limitation regarding performance on inorganic compounds and polymers (due to BRICS being designed for organic molecules) is acknowledged but not quantified. A small experiment or a more detailed discussion on the extent of this performance drop and potential mitigation strategies (e.g., adapting BRICS rules) would be valuable.

**Questions:**

1.Your work presents a compelling bimodal architecture. However, several other studies have proposed multi-modal models for molecules, such as MoleculeSTM (structure-text) and 3DToMolo (3D structure-text), which also emphasize rich, cross-modal alignment. Could you discuss how your descriptor-based language model approach compares philosophically and technically to these text-based multi-modal models? What are the relative advantages of using physicochemical descriptors over natural language text as a modality paired with structure?

2.You employ a contrastive loss to align the embeddings from the two modalities. Have you considered or experimented with more direct fusion mechanisms, such as using cross-attention where the graph model's output can attend to the language model's sequence of substructure embeddings (or vice versa)? Do you think such an approach could capture more fine-grained, substructure-to-graph-node relationships?

---

### Official Review · Reviewer_UetE · 2025-11-01

**Soundness:** 2
**Presentation:** 2
**Contribution:** 2
**Rating:** 4
**Confidence:** 4

**Summary:**

The paper introduces two models that combine RoBERTa with graph neural networks to improve molecular representation learning. One model integrates RoBERTa with a GCN, and the other with a GIN. During training, atom and bond features are masked, and contrastive learning is used to align the representations from text and graph views of the same molecule. The models are evaluated on QSAR tasks to assess their ability to predict molecular properties.

**Strengths:**

The method demonstrates strong performance on downstream molecular property prediction tasks.

**Weaknesses:**

Using “Thinking Like a Chemist” in your title naturally sets the expectation that the model can reason about chemistry, make decisions, or mimic a chemist’s problem-solving process (e.g., predicting reaction outcomes, designing new molecules intelligently, or explaining chemical phenomena). Your paper is about learning molecular representations --> the title is misleading, because representation learning alone does not involve reasoning
The approach of fragmenting molecules using BRICS and representing them with BRICS substructures plus additional substructures (Figure 2) does not preserve the attachment points between fragments. As a result, the original molecule cannot be reconstructed from these fragments, and distinct molecules may yield identical fragment sets. It is recommended to include a wildcard atom at the attachment points to retain connectivity information and better distinguish different molecules.
It is recommended to include an ablation study comparing model performance with and without the use of additional substructures, to clarify their contribution.
Minors Comments:
Code availability: Link shows “The requested file is not found.”
The manuscript would benefit from a more detailed explanation of QSAR, which is the main task.
Datasets such as QM7, QM8, QM9, FreeSolv, ESOL, and Lipo are more focused on molecular property prediction rather than classical QSAR tasks.
[L93] The phrase “relationships between molecular properties” is imprecise, Hansch et al. actually studied relationships between molecular structure and properties (not between properties themselves).
Figure 3 labels the output of RoBERTa as “Bert Embedding (768 dim)”. This is inaccurate terminology: RoBERTa is not BERT, even though it shares the architecture. Correct naming would be “RoBERTa embedding” or “Language model embedding”. Calling it “Bert Embedding” could confuse readers.
[L374] The sentence “By splitting into chemically relevant substructures, leveraging a more physics-based input format (descriptors) and employing one of the most sophisticated language models, we achieve a significant milestone: a language model (LM) trained from scratch with 10 million entries from the PubChem dataset” is confusing. It is unclear what is meant by “splitting into chemically relevant substructures” and how this, combined with using descriptors and a language model, leads to the claimed milestone. Additionally, the term “entries” is vague — if it refers to molecules, then training on 10 million molecules is not particularly novel, as prior models such as ChemBERTa and other frameworks have been trained from scratch on larger datasets. The authors should clarify what they mean by substructures, descriptors, and entries, and provide justification for why this constitutes a significant milestone.

**Questions:**

In figure 3, “Masked 15% tokens → RoBERTa → Language Modeling Head → CrossEntropyLoss” suggests manual masking before feeding tokens. RoBERTa can handle masking internally, so pre-masking may be redundant. Can you clarify if you are fine-tuning RoBERTa for custom token sequences derived from descriptors? If so, what is the reason for that?

---

### Official Review · Reviewer_RKcm · 2025-11-03

**Soundness:** 2
**Presentation:** 2
**Contribution:** 2
**Rating:** 2
**Confidence:** 4

**Summary:**

This paper proposes a bimodal framework combining language and graph-based molecular representations for chemical property prediction. Instead of relying on SMILES sequences, the authors decompose molecules into chemically meaningful substructures using the BRICS fragmentation algorithm and compute physicochemical descriptors for each fragment. These descriptor sequences are used as input tokens for a RoBERTa-based language model (“SubD-BERT”), which is jointly trained with a graph neural network (GIN, GCN, or Graphormer) through contrastive learning. The goal is to integrate local substructural information with global graph connectivity to improve downstream QSAR and property prediction tasks.

**Strengths:**

The paper presents a clear methodological pipeline that bridges chemical descriptors with transformer-based and graph-based architectures, emphasizing interpretability and domain knowledge. The BRICS-based decomposition is a reasonable choice for fragment-level modeling, offering a structured alternative to SMILES-based tokenization. The contrastive learning framework for aligning substructure- and graph-level embeddings is technically well-motivated. Performance improvements on several benchmarks suggest that the integration of fragment-level physicochemical information can enhance predictive accuracy.

**Weaknesses:**

Despite its clarity, the paper lacks true novelty and broader experimental support. The approach essentially reuses established components (RoBERTa, GCN/GIN/Graphormer, BRICS fragmentation, and descriptor-based features) and combines them without demonstrating a clear new principle or theoretical insight.
The claim of “thinking like a chemist” remains largely rhetorical—there is no evidence that the model captures reasoning-like processes, causal relations, or interpretable chemistry. Moreover, the rationale for using separate “substructure descriptor sequences” is not well justified: it’s unclear why modeling them as language tokens is necessary rather than directly integrating them within a graph-level embedding.
The experiments are restricted to small- and medium-scale datasets, lacking tests on more diverse molecular sets (e.g., large inorganic or polymeric systems) or realistic property prediction tasks. There is also a conceptual gap between the descriptor-level improvements and any higher-level chemical understanding.
Finally, the evaluation does not include comparisons to newer descriptor-aware or 3D-augmented multimodal baselines (e.g., MACE, Allegro, COMFormer, Matformer, Uni-Mol XL), limiting the validity of claims about outperforming “modern frameworks.” The most recent model is from 2023. Especially, graph neural networks are highly studied in this field and architectures such as FAENet, Equiformer v2 and similars should have been included instead of models from 2020.

**Questions:**

Why is the “language” modality needed at all? Could descriptor sequences not be integrated directly into the graph architecture as node or edge features?

How sensitive are the results to the choice of descriptor sets or the BRICS fragmentation scheme? Would random fragmentation yield comparable performance?

Does the model handle stereochemistry, tautomers, and chirality explicitly in descriptor encoding?

How does the proposed approach scale with larger datasets (e.g., 100M+ molecules) or more complex systems like metal-organic frameworks or polymers?

Could you show interpretability analyses or attention maps demonstrating that the model indeed “thinks like a chemist,” i.e., captures chemically meaningful substructures?

---

### Note · Authors · 2025-11-12

I have read and agree with the venue's withdrawal policy on behalf of myself and my co-authors.